# Development of an Objective Scoring System for Endoscopic Assessment of Radiation-Induced Upper Gastrointestinal Toxicity

**DOI:** 10.3390/cancers13092136

**Published:** 2021-04-29

**Authors:** Daniel Lin, Shalini Moningi, Joseph Abi Jaoude, Ben S. Singh, Irina M. Cazacu, Ramez Kouzy, Graciela M. Nogueras Gonzalez, Phonthep Angsuwatcharakon, Joseph M. Herman, Manoop S. Bhutani, Cullen M. Taniguchi

**Affiliations:** 1Department of Radiation Oncology, Division of Radiation Oncology, University of Texas MD Anderson Cancer Center, 1515 Holcombe Blvd, Houston, TX 77030, USA; DLin4@mdanderson.org (D.L.); JbAbi@mdanderson.org (J.A.J.); RKouzy@mdanderson.org (R.K.); 2Department of Radiation Oncology, Brigham and Women’s Hospital, 75 Francis St, Boston, MA 02115, USA; SMoningi@bwh.harvard.edu; 3Department of Gastroenterology, Hepatology and Nutrition, Division of Internal Medicine, University of Texas MD Anderson Cancer Center, 1515 Holcombe Blvd, Houston, TX 77030, USA; BSingh21@mdanderson.org; 4Research Center of Gastroenterology and Hepatology, University of Medicine and Pharmacy of Craiova, Strada Petru Rares 2, Craiova 200349, Romania; Irina.Cazacu@umfcv.ro; 5Department of Biostatistics, Division of Basic Sciences, University of Texas MD Anderson Cancer Center, Houston, TX 77030, USA; GNoguera@mdanderson.org; 6Department of Anatomy, Faculty of Medicine, Chulalongkorn University, Bangkok 10330, Thailand; Phonthep.A@chula.ac.th; 7Department of Radiation Medicine, Northwell Health Cancer Institute, 450 Lakeville Road, New Hyde Park, NY 11042, USA; JHerman1@northwell.edu

**Keywords:** pancreatic cancer, radiation injuries, upper gastrointestinal tract, endoscopy, radiosurgery

## Abstract

**Simple Summary:**

High-dose radiation therapy techniques have gained increasing interest in pancreatic cancer treatment, but toxicity to the upper gastrointestinal (GI) organs remains a major concern. We aimed to develop an objective toxicity scoring system to be used during endoscopic evaluation that allows for direct assessment of the stomach and duodenum before and after radiation treatment. Our toxicity scoring takes into account the pathological categories of erythema, edema, ulceration, and stricture to determine radiation-related GI toxicity. We assessed and validated the upper GI toxicity of 19 locally advanced pancreatic cancer trial patients undergoing stereotactic body radiation therapy (SBRT). With future usage, we hope this scoring system will provide objective and reliable assessments of changes in GI toxicity during the radiation treatment of pancreatic cancer and for GI toxicity assessment during radiation therapy research trials.

**Abstract:**

We developed and implemented an objective toxicity scoring system to be used during endoscopic evaluation of the upper gastrointestinal (GI) tract in order to directly assess changes in toxicity during the radiation treatment of pancreatic cancer. We assessed and validated the upper GI toxicity of 19 locally advanced pancreatic cancer trial patients undergoing stereotactic body radiation therapy (SBRT). Wilcoxon-signed rank tests were used to compare pre- and post-SBRT scores. Comparison of the toxicity scores measured before and after SBRT revealed a mild increase in toxicity in the stomach and duodenum (*p* < 0.005), with no cases of severe toxicity observed. Kappa and AC1 statistics analysis were used to evaluate interobserver agreement. Our toxicity scoring system was reliable in determining GI toxicity with a good overall interobserver agreement for pre-treatment scores (stomach, κ = 0.71, *p* < 0.005; duodenum, κ = 0.88, *p* < 0.005) and post-treatment scores (stomach, κ = 0.71, *p* < 0.005; duodenum, κ = 0.76, *p* < 0.005). The AC1 statistics yielded similar results. With future usage, we hope this scoring system will be a useful tool for objectively and reliably assessing changes in GI toxicity during the treatment of pancreatic cancer and for GI toxicity assessments and comparisons during radiation therapy research trials.

## 1. Introduction

Pancreatic cancer (PCA) remains a leading cause of cancer-related deaths in the United States, with a five-year overall survival of 8% [1]. PCA is potentially curable with surgery, but this is limited to less than 15% of all patients with this diagnosis. When surgery is not possible, local progression remains a major source of morbidity and mortality [2]. The use of chemoradiation is also common in PCA; however, no clear guidelines exist on what regimen to use, owing to inconclusive and contradictory data from previous literature. Furthermore, chemoradiation has only modest benefits, in part because the dose given to the pancreas is limited by toxicity to the nearby duodenum [3]. Stereotactic body radiation therapy (SBRT) has gained increased interest in PCA treatment with advanced image guidance to limit duodenal dosing [4,5]. However, severe toxicity can still be a problem, especially when using a dosage that is considered ablative [6,7,8].

While most agree that the bowel carries the greatest risk of toxicity after chemoradiation for pancreatic cancer, assessing the toxicity of the gastrointestinal (GI) tract has proven to be inconsistent and difficult. The Common Terminology Criteria for Adverse Events (CTCAE) clinical measurements of toxicity are often used, but these can provide an incomplete and subjective assessment of radiation-induced GI toxicity due to the nature of physician reporting, limited categories, and potential misuse [9,10]. Additionally, patients may only report toxicity to physicians when it is severe, which may limit the ability to further improve radiation treatments using these data [11,12]. Other methods of reporting toxicity, such as patient-reported outcomes (PROs) remain scarce due to their complexity of design and implementation [13,14]. These issues lead to a lack of consensus on how to monitor for toxicity during pancreatic cancer treatment. Direct endoscopic visual assessment of the GI tract surrounding the target site before and after radiation treatment will allow us to better characterize the toxicity of SBRT during PCA treatment and any future treatments that may impact the GI tract.

To better understand the early toxicity after SBRT for PCA, we developed an endoscopic scoring system to evaluate and document the objective parameters of GI toxicity after radiation therapy. The objective scoring system was developed to evaluate erythema, edema, ulceration, and stricture of the duodenum and stomach during EGD (esophagogastroduodenoscopy) before and after radiation treatment. We piloted this system as part of a prospective trial using dose-escalated SBRT for PCA, where we might expect to find higher levels of toxicity. Furthermore, we found that our endoscopic scoring system is facile and can be independently reproduced by other endoscopists.

In this study, we included patients with locally advanced pancreatic cancer (LAPC) receiving endoscopic ultrasound (EUS)-guided fiducial marker placement and high-dose SBRT. The primary outcome is to assess acute toxicity within 90 days. Consequently, we aimed to test the reliability of the scoring system in a cohort of patients undergoing SBRT for LAPC.

## 2. Results

### 2.1. Patient Characteristics

Table 1 presents an overview of patient characteristics for the 19 patients receiving high-dose SBRT included in the study. The study included nine males and 10 females with a median age of 71 years. All patients were assessed for toxicity during endoscopy before SBRT, but only 17 patients received a post-SBRT endoscopic assessment. All patient tumor histology revealed adenocarcinoma of the pancreas. Eight patients (42%) had tumors located in the head of the pancreas, nine (47%) had tumors located in the body, and two (11%) had tumors localized in the tail.

### 2.2. Toxicity Scoring

Table 2 presents a summary of the pre- and post-SBRT toxicity scores for both the duodenum and stomach. Pre-SBRT toxicity was assessed for all 19 patients. Both the duodenum and stomach were assessed. Of those 19 patients, 17 patients were assessed for post-SBRT toxicity scores as two of the patients were unable to undergo post-SBRT endoscopy assessment. The median pre-SBRT toxicity score was 0, indicating no toxicity, for both the duodenum and stomach. The median post-SBRT score was 1, indicating mild toxicity, for both the duodenum and stomach. The median change in toxicity from pre- to post-SBRT was one point for both the stomach and duodenum. The pre-SBRT total toxicity scores were significantly different from the post-SBRT total toxicity scores (*p* < 0.005) for both the duodenum and stomach assessments. Further analysis was performed to assess differences in the individual toxicity characteristics for pre- and post-SBRT. Erythema scores for before and after SBRT treatment for the stomach and duodenum were significantly different (*p* < 0.005). Analysis of the pre- and post-SBRT edema scores yielded *p* < 0.05 for the duodenum and no significant change for the stomach. For both the ulcer score and the stricture score, no significant difference was found when comparing the pre- and post-SBRT scores.

### 2.3. Reliability Testing

Table 3 summarizes the results of interobserver agreement or reliability testing with two unique raters using kappa-statistics for the objective parameters of the radiation-induced GI toxicity. The percentage of agreement reached 100% for two objective parameters, ulcers and strictures (κ = 1). There was a good overall agreement for the pre-treatment scores for both the duodenum and stomach (κ = 0.71, *p* < 0.005 and κ = 0.88, *p* < 0.005, respectively). Similarly, there was a good interobserver agreement for overall post-treatment scores (stomach, κ = 0.71 and duodenum, κ = 0.76, *p* < 0.005). To further validate our results, we also performed an interobserver analysis using Gwet’s AC1 (Table 3). Overall, the results were similar and in line with the kappa analysis.

### 2.4. Outcomes

Of the 19 patients, eight patients exhibited only distant progression, one patient exhibited only local progression, and four patients exhibited both distant and local progression. The total median follow-up time was eight months.

## 3. Discussion

In this study, we reported a new toxicity scoring system that can help evaluate toxicity of the duodenum and stomach prior to and following the delivery of high doses of radiation therapy via SBRT. We found low rates of local toxicity following the delivery of SBRT to locally advanced pancreatic tumors. The majority of patients (12 of 17 patients) had mild aggregate endoscopic toxicity scores (≤2) for both the stomach and duodenum following SBRT and, even more importantly, 13 of 17 patients exhibited a low increase in toxicity (≤2) from pre- to post-SBRT for both organs. Additionally, reliability analysis of our toxicity scoring system revealed the potential to not only apply this scoring system to future endoscopy cases requiring objective toxicity assessment, but to past endoscopic cases as well through imaging.

The current standard of care for patients with LAPC includes a combination of chemotherapy and radiation therapy. Over the last decade, technological advances and research have resulted in significant improvements in the quality of radiation and imaging and a better ability to deliver higher radiation doses more accurately and precisely. SBRT is an emerging treatment option for patients with locally advanced disease. There have been many successful clinical trials and retrospective data published on the safety and efficacy of SBRT [15,16,17]. SBRT has been shown to provide excellent local control benefits due to higher doses delivered to the tumor and decreased toxicity to the surrounding organs by targeting a smaller area. Additionally, SBRT is usually delivered in three to five fractions (one fraction delivered per day) and allows for less time off of systemic therapy. This is particularly important for PCA, given the majority of patients with this illness fail distantly, which possibly explains the improved overall survival of SBRT patients when compared to longer courses of radiotherapy [18,19,20]. As part of our institution’s dose-escalation trial, LAPC patients undergo a five-fraction regimen of SBRT over five consecutive days, consisting of a total of 50 or 55 Gy to the pancreatic tumor and tumor vessel interface.

Presently, there are no studies that have endoscopically evaluated patients post-SBRT for radiotherapy-related toxicity and local tumor progression. Additionally, a standardized way to grade duodenal and gastric toxicity from SBRT has yet to be established [21]. Consequently, we developed and implemented a scoring system to objectively assess radiation-induced GI toxicity during an upper endoscopy. At 12 weeks following the completion of SBRT, our patients receive an endoscopy for toxicity evaluation. This post-SBRT endoscopy allows us to further assess toxicity in the duodenum and stomach and can, at times, be more accurate than reviewing post-SBRT CT scans. A follow-up endoscopy could also assess local tumor progression in the duodenum or stomach. In this study, we found reasonable rates of toxicity, with no severe toxicity seen on endoscopic evaluation. Our results showed that SBRT delivery to doses up to 55 Gy in five days is safe and well-tolerated from a GI toxicity standpoint.

Our study demonstrated that our toxicity score is reliable for assessing GI toxicity, with good interobserver agreement for all of the objective parameters evaluated. To assess the validity of a scoring system, the relationship between the objective findings reported by the observers should be correlated with the subjective complaints of the patients. However, for pancreatic cancer patients, subjective complaints, such as abdominal pain, nausea, and vomiting, may occur as a result of chemotherapy or local progression of the disease and not as a result of radiation. As a result, validity testing was not applicable and was thus not performed in the present study.

It is worth noting that the toxicity system used in this study was part of an exploratory aim of our research. As such, we did not alter any clinical practice based on the results of this study, and all patients were treated routinely with proton pump inhibitors and adequate diet modifications. As this scoring system becomes more studied and validated in the future, it could be used to alter prophylactic or therapeutic treatment in patients receiving radiation therapy to pancreatic tumors.

The major limitation of this study is our small sample size, with only 19 patients included and recruited at a single institution. Our study was designed as a pilot study that needs further validation in the future. Moreover, all endoscopy procedures were performed and initially graded by the same endoscopist, although reliability was later validated by a separate endoscopist. Furthermore, only two non-blinded endoscopists were responsible for rating toxicity, and the second endoscopist only had access to select images taken. As such, this could have introduced some bias into our results. We hope that other groups will apply this simple objective scoring system when appropriate and provide external validation of its reproducibility and utility. Since post-SBRT endoscopic evaluation is not standard of care, this procedure would not be easy to perform routinely and would require extensive research funding. As such, we hope that ongoing or future trials implement our scoring system when possible, in order to get a larger sample size that would better assess the scoring validity. There are no restrictions, limitations, or licensing requirements for its use, and it is freely accessible to colleagues in the field.

## 4. Materials and Methods

### 4.1. Trial Description

The 19 patients included in this study were treated in an institutional IRB-approved trial at a single institution (ClinicalTrials.gov Identifier: NCT03340974). This trial is an adaptive phase I/II dose-escalation trial looking at the delivery of escalated doses of SBRT for patients with LAPC. The trial randomized patients to either a placebo (control arm) or the drug (experimental arm), which is a compound with the potential to promote radioprotection of the stomach and duodenum from high doses of RT. The trial arms are double-blinded and, at the time of this manuscript (June 2020), the assignment of a placebo or the drug is unknown, while the doses of radiation are known. All patients received a minimum of three months of standard chemotherapy, which consisted of FOLFIRINOX, gemcitabine + *nab*-paclitaxel, or a sequential combination of both. The SBRT dose on the placebo and experimental arms begin at level I, which was 50 Gy/5 fractions. Escalation to dose levels II (55 Gy/5 fractions) or level III (60 Gy/5 fractions) was determined by LO-ET methodology [22]. The trial is ongoing, but at the time of this analysis (May 2020), the average length of chemotherapy prior to SBRT was five months. Twelve patients were treated with dose level I and seven were treated with dose level II.

### 4.2. Endoscopy Description/Fiducial Placement

After obtaining informed consent and enrollment in the trial, patients underwent an EGD and subsequent EUS-guided fiducial marker placement. Gold fiducial markers (0.28 × 20 mm in size) were placed in order to visualize the tumors during radiotherapy treatment planning and delivery. During the upper endoscopy procedure, the stomach and duodenum were assessed for any abnormalities or toxicities. At the 12-week follow-up post-SBRT, patients underwent a repeat endoscopy, where the stomach and duodenum were assessed for radiotherapy-related toxicity.

### 4.3. Toxicity Scoring System

An upper endoscopy allows for visual assessment of the stomach and duodenum. Radiation-induced GI toxicity was directly evaluated by the endoscopist. Appendix A shows an example of the score sheet that was used during endoscopy before radiation treatment and during the follow-up endoscopy post-radiation. This scoring system was developed in a multidisciplinary fashion, with discussion between an experienced endoscopist (M.S.B.) and two experienced radiation oncologists (C.M.T. and J.M.H.). For ease of use and future reference, we titled this scoring system the BTH Scoring System. The toxicity severity was quantitatively recorded using scores of 0 (no toxicity), 1 (mild toxicity), or 2 (moderate/severe toxicity). The endoscopist assigned these scores to the following four categories assessing pathological findings: Erythema, edema, ulceration, and stricture. If ulcers were present, the degree of ulceration was recorded noting the cumulative estimated surface area and the stigmata of hemorrhage. The sum of the categorical scores was used to determine the aggregate toxicity score and to assign a qualitative descriptor representing overall toxicity: No toxicity (0), mild (1–2), moderate (3–4), or severe (≥5). A general overview and guide to the toxicity scoring system, as well as examples of toxicity, are shown in Figure 1, Figure 2 and Figure 3.

### 4.4. Statistical Methods

Descriptive statistics were generated for all patients (*n* = 19). Statistical comparisons of pre- and post-SBRT toxicity scores (*n* = 17) were assessed by the Wilcoxon signed-rank test using GraphPad Prism version 8.0.0 for macOS (GraphPad Software, San Diego, CA, USA). Statistical analysis for reliability was performed using Stata/SE version 15.1 statistical software (Stata Corp. LP, College Station, TX, USA).

### 4.5. Inter-Rater Reliability Testing and Validation

The reliability of the scoring system was investigated by calculating the percent agreement between two observers, an experienced endoscopist (M.S.B., 28 years of experience) and an advanced endoscopy fellow (P.A., 10 years of experience) for both the objective findings. Scores were actively assigned by one endoscopist during the initial endoscopy, while the other endoscopist retroactively and independently assigned scores based on endoscopic imaging of the same cases. Kappa statistics and Gwet’s AC1 analyses were performed to evaluate interobserver agreement and to consider situations in which the agreement may be due to chance.

## 5. Conclusions

The direct and objective measurements of GI toxicity effects of treatments will provide valuable information regarding the safety and consequences of future treatments that may be missed by other methods of evaluation. We conclude that endoscopic evaluation pre- and post-SBRT treatment for pancreatic cancer is safe and effective for measuring treatment-related upper GI toxicity. Upper endoscopy supplemented with this scoring system could be utilized more often for LAPC patients following the completion of SBRT treatment to objectively assess upper GI toxicity and to determine the potential for impending complications (e.g., bleeding, perforation, or obstruction) or for the formal assessment of radiation-induced upper GI toxicity in research studies comparing the benefits and harms of different treatment arms.

## Figures and Tables

**Figure 1 cancers-13-02136-f001:**
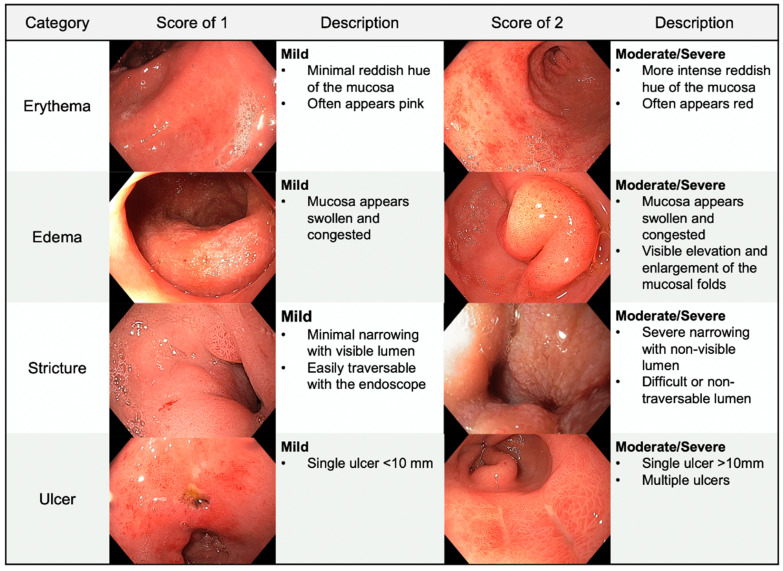
Endoscopic toxicity scoring categories. The above images provide examples of our toxicity scores and the corresponding descriptions for erythema, edema, stricture, and ulcer.

**Figure 2 cancers-13-02136-f002:**
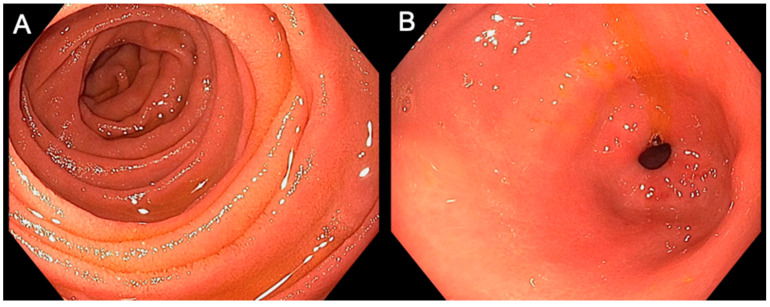
Examples of normal gastric and duodenal mucosa. The images above were taken during endoscopic examination of a patient’s duodenum (**A**) and gastric antrum (**B**). There was no visual evidence of any toxicity characteristics, indicating no toxicity for both organs. Of note, this patient was not a part of this study cohort.

**Figure 3 cancers-13-02136-f003:**
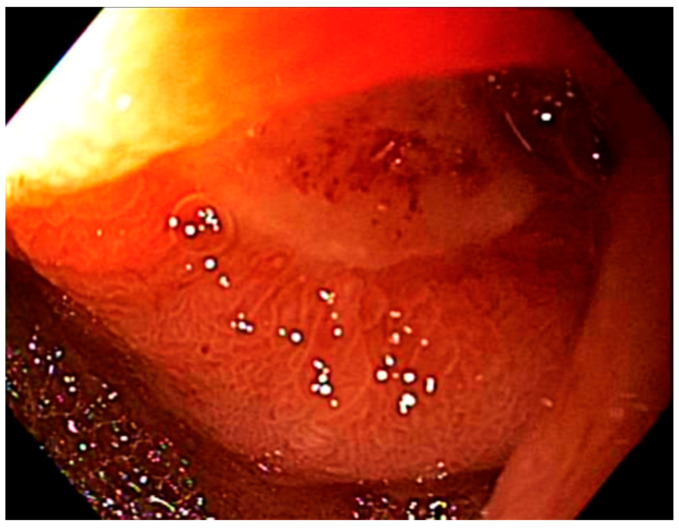
Severe toxicity in the duodenum. This image was taken during endoscopic examination of a patient’s duodenum. This patient was not a part of this study cohort. Presence of a single ulcer with severe stricture and moderate edema dictates a total toxicity score of 5, indicating severe duodenal toxicity.

**Table 1 cancers-13-02136-t001:** Study patient demographics.

Characteristic	Patients
Age (*n* = 19)	
Mean ± SD	68.6 ± 11.2
Median (IQR)	71 (64–76)
Sex—no. (%)	
Male	9 (47.4)
Female	10 (52.6)
Baseline ECOG—no. (%)	
0	8 (42.1)
1	11 (57.9)
T stage—no. (%)	
T4	19 (100)
N stage—no. (%)	
Nx	2 (10.5)
N0	14 (73.7)
N1	3 (15.8)
M stage—no. (%)	
M0	19 (100)
SBRT dosage—no. (%)	
50 Gy	12 (63.2)
55 Gy	7 (36.8)
Chemotherapy—no. (%) *	
Gemcitabine and abraxane	11 (57.9)
Folfirinox	13 (68.4)
Tumor location—no. (%)	
Head	8 (42.1)
Body	9 (47.4)
Tail	2 (10.5)
Baseline CA 19-9 (*n* = 18)	
Mean ± SD	178.2 ± 513.9
Median (IQR)	29.6 (8–60.4)

* Some patients received multiple types of chemotherapy.

**Table 2 cancers-13-02136-t002:** Endoscopic toxicity grades of pancreatic cancer patients undergoing stereotactic body radiation therapy (SBRT).

Characteristic	None (0) No. (%)	Mild (1–2) No. (%)	Moderate (3–4) No. (%)	Severe (≥5) No. (%)
Duodenal Toxicity				
Pre-treatment (*n* = 19)	15 (79)	4 (21)	0 (0)	0 (0)
Post-treatment (*n* = 17)	5 (29)	8 (47)	4 (24)	0 (0)
*Δ Toxicity (*n* = 17)	8 (47)	5 (29)	4 (24)	0 (0)
Gastric Toxicity				
Pre-treatment (*n* = 19)	10 (53)	9 (47)	0 (0)	0 (0)
Post-treatment (*n* = 17)	3 (18)	11 (64)	3 (18)	0 (0)
*Δ Toxicity (*n* = 17)	6 (35)	10 (59)	1 (6)	0 (0)

*Δ difference in toxicity calculated from pre- to post-treatment.

**Table 3 cancers-13-02136-t003:** Toxicity scoring inter-rater agreement.

Characteristic (*n* = 17)	Agreement	Kappa	*p*-Value	Gwet’s AC1	*p*-Value
Pre-treatment stomach toxicity					
Erythema	0.71	0.41	0.046	0.42	0.085
Edema	0.94	0.77	0.001	0.92	<0.001
Total	0.71	0.49	0.003	0.59	0.002
Pre-treatment duodenum toxicity					
Erythema	0.94	0.82	<0.001	0.91	<0.001
Edema	0.94	0	-	0.94	<0.001
Total	0.88	0.65	0.001	0.86	<0.001
Post-treatment stomach toxicity					
Erythema	0.76	0.62	<0.001	0.66	0.001
Edema	0.82	0.63	0.001	0.77	<0.001
Total	0.71	0.6	<0.001	0.64	<0.001
Post-treatment duodenum toxicity					
Erythema	0.82	0.71	<0.001	0.75	<0.001
Edema	0.82	0.67	0.001	0.76	<0.001
Ulcers	1.00	1.00	<0.001	1.00	-
Strictures	1.00	1.00	<0.001	1.00	-
Total	0.76	0.7	<0.001		

## Data Availability

All data are available upon reasonable request to the corresponding authors.

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
