# Peer review of "Development of an Objective Scoring System for Endoscopic Assessment of Radiation-Induced Upper Gastrointestinal Toxicity"

_cancers, 2021, doi:10.3390/cancers13092136_

Round 1

Reviewer 1 Report

I thanks the authors for addressing all my comments in a satisfactory fashion.  I would still encourage adding (even as a supplementary table) any data regarding clinical correlation between the endoscopic toxicity and clinical symptoms - which will improve the clinical ability to evaluate for possible use of this endoscopic score in actual managing of patients.  

Reviewer 2 Report

I have no further comments.

Reviewer 3 Report

Lin and colleagues present their revised manuscript suggesting a new endoscopic scoring system to predict gastro-intestinal toxicity following stereotactic radiotherapy for pancreatic cancer. The authors demonstrated a good overall interobserver agreement of the model, and suggest future validation of the system to further refine therapy. The authors adequately described the limitations of the study, mainly the small sample-size and a monocenter study design. I think the paper is well-written and interesting given the increasing applicability of radiotherapy in the management of pancreatic cancer. I have no further comments to improve the paper and recommend publication.

This manuscript is a resubmission of an earlier submission. The following is a list of the peer review reports and author responses from that submission.

Round 1

Reviewer 1 Report

I congratulate the author for their interesting and thought-provoking approach to objectively address and quantify organ cytotoxicity.

I have a comments to share regarding this study:

Introduction -

It is important to note that the data on Chemoradiation or just upfront SBRT in LAPC in still inconclusive as there are multiple studies published with contradictory results.

Are the authors proposing this just for PCA or for assessment of chemotherapy for other GI cancers?

Was a colonoscopy considered to assess colonic toxicity as well?

Methodology -

This study is limited in it's generalizability by the fact that there are only two raters? Additionally if the second rater had only access to the select images acquired during the exam and not to the full video - that creates a high probability of bias.

Clinical correlates - How does this objective grading scale correlate with the CTCAE reporting system?

Results-

Table 1 -Under chemotherapy the total amount of patient is 24 (11+13) - is that an error?

Reviewer 2 Report

This is an interesting paper investigating the possibility to evaluate endoscopically the toxicity of SBRT for locally advanced PDAC. Overall the methodology is correct and clear, the manuscript well written and comprehensive.

To me, there three major points. First, the number of patients evaluated is small, with only 17 patients undergo endoscopy evaluation before and after SBRT. Second, endoscopic evaluations were performed by the same endoscopist, who were not blinded to the study. This creates a bias during score assignment. Of course, the interobserver agreement could overcome this limit, but here we came to the third point. Only two raters were involved in the interobserver agreement and no cases of severe toxicity were present, thus weakening the interobserver agreement analysis. These points limit the validity of the conclusions of this study. At least 40 cases should be enrolled and 5/6 raters involved for a strong interobserver agreement.

This study has great potential. I suggest: 1) to increase the number of patients enrolled (a formal sample size calculation is not possible, but at least 40 patients should be included); 2) to implement the interobserver agreement.

Minor:

- Endoscopic videos (and not images) should be used for the interobserver agreement. 

- Besides the K statistic, I suggest to use also the Gwet’s AC1. Indeed, the kappa statistic is affected by the prevalence of the finding under consideration to a similar extent as predictive values are affected by the prevalence of the considered disease. For rare findings, very low values of kappa may not necessarily reflect low rates of overall agreement.18,19The Gwet measure AC1 is supposed to deal with the apparent “paradox” of low agreement values despite a large percentage agreement

Reviewer 3 Report

In their manuscript entitled "Development of an Objective Scoring System for Endoscopic Assessment of Radiation-Induced Upper Gastrointestinal Toxicity" the authors present the outcomes of an original prospective study which sought to   develope an objective toxicity scoring system to be used during endoscopic evaluation that allows for direct assessment of the stomach and duodenum before and after radiation treatment.The authors assessed and validated upper GI toxicity of 19 LAPC trial patients undergoing SBRT. Based on their outcomes the authors conclude that their toxicity scoring system was reliable in determining GI toxicity with a good overall interobserver agreement for pre- scores and post-treatment scores. 

This is a very interesting study focusing on a significant subset of patients with unresectable LAPC, which are increasingly led to radiation treatment with SBRT, which however is associated with significant toxicity rates. The title accurately reflects the subject of the manuscript. The authors adequately discuss the background of the study. The materials and methods as well as the results are sufficiently described and clear to the readership. The sample size of included patients is indeed small yet such is acknowledged by the authors in their limitations section. The discussion section is optimal in length and all current literature in context is presented. All references are current. Figures are highly illustrative and helpful to the readership.

Reviewer 4 Report

The authors present a sub-study of a pending randomized-controlled trial, comparing stereotactic body-radiation therapy (SBRT) to placebo for locally advanced pancreatic cancer (LAPC). In the present study, the authors aimed to develop a toxicity scoring system to determine toxicity of the gastro-intestinal tract after SBRT for LAPC upon endoscopic evaluation.

I believe that the study is well-written and has some merit, but it must be noted that the study has certain methodological flaws. My comments are listed below.

Major:

  • As also stated by the authors, the main limitation of the present study is a small sample-size (n=19), within a mono-center setting. The findings of the present study were not validated in another center or by a different endoscopist. For this reasons, the study lacks external validity. It would greatly improve the paper if the results could be externally validated.
  • The authors could elaborate more on the clinical importance of the scoring system. Were patients with higher grade toxicity scores upon endoscopic evaluation treated differently than in clinical practice?
  • It would be interesting if the authors could correlate the toxicity scores with actual patient-reported outcomes. This seems to be a more relevant primary outcome than interobserver agreement.

Minor:

  • Preferably, sentences are not started with numbers, and numbers <10 could be written out in full.
  • Experience years of the endoscopist could be added.